# Comparative In Vitro Toxicology of Novel Cytoprotective Short-Chain Naphthoquinones

**DOI:** 10.3390/ph13080184

**Published:** 2020-08-07

**Authors:** Zikai Feng, Mohammed Sedeeq, Abraham Daniel, Monika Corban, Krystel L. Woolley, Ryan Condie, Iman Azimi, Jason A. Smith, Nuri Gueven

**Affiliations:** 1School of Pharmacy and Pharmacology, University of Tasmania, Hobart, TAS 7005, Australia; mohammed.sedeeq@utas.edu.au (M.S.); abraham.daniel@utas.edu.au (A.D.); monika.corban@utas.edu.au (M.C.); iman.azimi@utas.edu.au (I.A.); 2School of Natural Sciences, University of Tasmania, Hobart, TAS 7005, Australia; krystel.woolley@utas.edu.au (K.L.W.); ryan.condie@utas.edu.au (R.C.); jason.smith@utas.edu.au (J.A.S.)

**Keywords:** mitochondria, short-chain quinone, idebenone, cytotoxicity

## Abstract

Short-chain quinones (SCQs) have been identified as potential drug candidates against mitochondrial dysfunction, which largely depends on the reversible redox characteristics of the active quinone core. We recently identified 11 naphthoquinone derivatives, **1**–**11,** from a library of SCQs that demonstrated enhanced cytoprotection and improved metabolic stability compared to the clinically used benzoquinone idebenone. Since the toxicity properties of our promising SCQs were unknown, this study developed multiplex methods and generated detailed toxicity profiles from 11 endpoint measurements using the human hepatocarcinoma cell line HepG2. Overall, the toxicity profiles were largely comparable across different assays, with simple standard assays showing increased sensitivity compared to commercial toxicity assays. Within the 11 naphthoquinones tested, the *L*-phenylalanine derivative **4** consistently demonstrated the lowest toxicity across all assays. The results of this study not only provide useful information about the toxicity features of SCQs but will also enable the progression of the most promising drug candidates towards their clinical use.

## 1. Introduction

Mitochondrial dysfunction causes a large number of diverse mitochondrial diseases, such as Friedreich’s ataxia (FA) [1], Leigh syndrome (LS) [2], mitochondrial encephalopathy, lactic acidosis, stroke-like episodes syndrome (MELAS) [3], maternally inherited diabetes and deafness (MIDD) [4], Leber’s hereditary optic neuropathy (LHON) [5], and dominant optic atrophy (DOA) [6]. Mitochondrial dysfunction is also described for many common inflammatory (i.e., ulcerative colitis) [7], neurodegenerative (i.e., Alzheimer’s disease, Parkinson’s disease, glaucoma, age-related macular degeneration) [8], neuromuscular (i.e., Duchenne muscular dystrophy (DMD), multiple sclerosis) [9], and metabolic disorders (i.e., diabetes, obesity) [10]. However, despite the high incidence of disorders with a mitochondrial pathology, there is a scarcity of approved drugs that aim to directly protect against mitochondrial dysfunction. This significant unmet medical need requires new drug candidates that could be of benefit to a multitude of indications. Potential drug candidates that protect against mitochondrial dysfunction include short-chain quinones (SCQs), which possess reversible redox characteristics due to their quinone core [11,12,13]. Several SCQs are currently in clinical development. The vitamin E derivative vatiquinone (EPI-743/PTC-743), an antioxidant that targets NAD(P)H:quinone oxidoreductase 1 (NQO1), was initially developed for FA (Phase II, NCT01962363, 3 × 400 mg for 18 months, with reported improved neurological functions) [14] and LS (Phase II, NCT01721733, 3 × 100 mg for 6 months, with reported improved movement) [15], and was recently acquired by PTC Therapeutics. Another vitamin E derivative, sonlicromanol (KH176, Khondrion), a reactive oxygen species (ROS) modulator, is in development for MELAS and MIDD (Phase II, NCT02909400, 2 × 100 mg for 28 days, with reported tolerance and safety) [16], LS and LHON (Phase I, NCT02544217, 800 mg for 7 days, with reported tolerance) [17]. The only drug currently available to patients is benzoquinone idebenone, which protects against vision loss and has even restored visual acuity in some LHON patients [18]. Especially for the subgroup of recently affected patients, idebenone has been shown to improve visual acuity and color vision [19,20]. A recent report also suggested that idebenone ameliorated mitochondrial complex I deficiency and stabilized/restored visual acuity in patients with DOA [6,21]. In contrast, earlier phase III clinical trials (NCT00905268; NCT00537680) in FA patients were unable to demonstrate therapeutic efficacy for idebenone [22]. Based on its cytoprotective activity under the conditions of mitochondrial dysfunction (hereafter referred to as mito-protection), idebenone was suggested for a wide range of disorders. Contrary to the widespread belief that idebenone is a CoQ_10_ analogue and acts as antioxidant, recent reports paint a very different picture: idebenone was reported to directly bind and inhibit p52Shc at nanomolar concentration [23], but also acts as PPARα/γ agonist, albeit at higher concentrations [24]. Finally, idebenone activates the expression of Lin28A in vivo, which was shown to be required for retinal neuroprotection and recovery of vision [25]. Although it is unclear at present if these activities of idebenone are causally connected, they all converge to activate Akt signalling, which alters metabolic functions, increases insulin sensitivity, increases mitochondrial function and stress resistance, and induces tissue repair. Although idebenone has consistently demonstrated very good safety in healthy subjects (2250 mg/day, 14 days) [26] and different patient groups (LHON patients: 900 mg/day, 24 weeks [18]; DMD patients: 900 mg/day, 52 weeks [27]), its efficacy is restricted by its limited absorption, a rapid first-pass effect [28], and its reliance on a single reductase for its bioactivation [12,13].

We recently reported the design and synthesis of a library of > 148 novel short-chain naphthoquinone derivatives [29] to overcome the known limitations of idebenone, such as limited bioactivation and rapid metabolic inactivation. From this library, 11 compounds (**1**–**11**, Table 1) showed significantly improved cytoprotective activity under the conditions of mitochondrial dysfunction and increased metabolic stability in vitro compared to idebenone [29,30]. The current study aimed to compare the in vitro toxicity of these 11 compounds against idebenone to identify possible drug candidates that could be progressed towards clinical development.

## 2. Results

### 2.1. WST-1 Assay

This study aimed to assess the in vitro toxicity of our test compounds in the hepatic cell line HepG2 across a range of assays to measure different toxicity-related endpoints including metabolic toxicity, membrane integrity, mitochondrial toxicity, mechanisms of cell death, DNA damage, and transformation potential. When cellular NAD(P)H synthesis as a surrogate marker for cellular metabolism was measured using the widely employed WST-1 dye, the reference compound idebenone significantly reduced WST-1 absorption from 150 μM onwards (*p* < 0.001; Figure 1a). In comparison, most test compounds already reduced absorption from 25 μM (**3** and **8**), 50 μM (**1**, **2**, and **9**–**11**), and 75 μM (**6** and **7**) onwards, respectively (*p* < 0.033; see Appendix A for full dataset). In contrast, compared to the untreated control cells, a significant reduction of WST-1 absorption by compounds **4** and **5** was only observed from 175 μM onwards (*p* < 0.002) and 200 μM (*p* < 0.001), respectively. Compared to idebenone, compounds **4** and **5** from 150 μM onwards reduced WST-1 absorption significantly less (*p* < 0.001).

### 2.2. ATP Levels

As another marker of metabolic toxicity, cellular ATP levels were assessed in the absence or presence of the reference or test compounds. The reference compound idebenone significantly reduced ATP levels from 125 μM onwards (*p* < 0.001; Figure 1b, Appendix A). Similar to the WST-1 results, a significant reduction was only observed at 200 µM by compound **4** and from 150 μM onwards by compound **5** (*p* < 0.033), respectively, while other compounds already significantly reduced ATP levels from 50–75 μM compared to the untreated cells (*p* < 0.033; Appendix A). Compared to idebenone, compounds **4** and **5** showed a significantly lower effect on ATP levels (*p* < 0.001) from 150 and 125 μM onwards, respectively (Figure 1b; Appendix A).

### 2.3. Protein Levels

Since our test compounds had similar effects on both ATP levels and the conversion of WST-1 dye, we assessed the levels of protein per well as a surrogate marker for cellular content. The reference compound idebenone significantly reduced protein levels from 100 μM onwards (*p* < 0.001; Figure 1c). Of 11 tested compounds, **4** and **5** significantly reduced protein levels from 100 μM onwards (*p* < 0.002), while significant reductions by the other compounds were already evident at 50–75 μM (*p* < 0.001; Appendix A). At higher concentrations from 125 μM onwards, compounds **4** and **5** affected protein levels significantly less than idebenone (*p* < 0.001).

### 2.4. Membrane Integrity

Propidium iodide (PI) staining was employed as a measurement of impaired membrane integrity. The reference compound idebenone significantly increased PI incorporation from 100 μM onwards (*p* < 0.033; Figure 2a). Of 11 tested compounds, **4** did not increase PI incorporation significantly at all tested concentrations, while 2 compounds increased from 150 μM (**5**, *p* < 0.033; **11**, *p* < 0.001), 1 compound from 125 μM (**8**, *p* < 0.001), 1 compound from 100 μM (**7**, *p* < 0.001), 4 compounds from 75 μM (**3**, **6**, **9** and **10**, *p* < 0.001), and 2 compounds from 50 μM onwards (**1**, *p* < 0.033; **2**, *p* < 0.001; Appendix A). Compared to idebenone, the observed effects by compounds **4**, **5**, and **11** was significantly lower from 100 μM onwards, at 125–150 μM, and at 100–125 μM, respectively (*p* < 0.033; Appendix A), while the effect by compound **8** was not statistically significant, and the effects by the other derivatives were significantly higher between 50 and 200 μM.

### 2.5. Multi-Tox Fluor Assay

Based on the effects of the test compounds on PI staining, we assessed cell membrane integrity using a commercially available kit that proposed to simultaneously assess this endpoint and cell viability. The reference compound idebenone did not significantly increase bis-AAF-R110 fluorescence, which is indicative of a lack of necrotic-cell protease activity at all test concentrations (Figure 2b). Of the 11 tested compounds, 5 compounds did not significantly increase bis-AAF-R110 fluorescence at any concentration (**1**–**5**), while compound **10** increased fluorescence at 200 μM (*p* < 0.033), compound **6** increased at 125–150 μM (*p* < 0.033), and four other compounds increased fluorescence from 125 μM onwards (**7**–**9** and **11**, *p* < 0.001; Appendix A). Compared to the effects of idebenone, no significant increases by compounds **1**–**6** were observed, while compound **10** showed significantly higher levels of bis-AAF-R110 fluorescence at 200 μM (*p* < 0.033), 3 compounds from 150 μM onwards (**8** and **9**, *p* < 0.002; **11**, *p* < 0.033), and compound **7** from 125 μM onwards (*p* < 0.033; Appendix A), respectively.

While bis-AAF-R110 measures the activity of the necrosis-associated protease, the protease substrate GF-AFC is thought to measure live cells with intact plasma membrane. In our test system, idebenone significantly reduced GF-AFC fluorescence from 75 μM onwards (*p* < 0.001; Figure 2c, Appendix A). In contrast, a significant reduction by compounds **4**, **5**, and **11** were only observed from 150, 125, and 175 μM onwards (*p* < 0.033), respectively. However, the remaining 8 compounds significantly reduced fluorescence already at 25–50 μM (*p* < 0.033; Appendix A). Compared to idebenone, a significantly lower reduction of fluorescence was detected for compounds **4**, **5**, and **11** from 75 μM onwards (*p* < 0.001, Appendix A).

### 2.6. Mitochondrial Superoxide Production

To further assess if the observed toxicity of the test compounds could be attributed to mitochondrial toxicity, mitochondrial superoxide production was measured. Antimycin A, used as a positive control in our test system [31], significantly increased the fluorescence of the mitochondrial superoxide indicator MitoSOX from 25 μM onwards (*p* < 0.001, Appendix A), while idebenone did not increase MitoSOX fluorescence across all tested concentrations (Figure 3). Of the 11 test compounds, 3 compounds significantly increased MitoSOX fluorescence to different degrees (**5**, 200 μM, *p* < 0.033; **8**, ≥100 μM, *p* < 0.033; **11**, ≥100 μM, *p* < 0.002), while for the 8 other test compounds no significant increases were detected. The observed increases by the 3 test compounds (**5**, **8**, and **11**) were significantly lower compared to antimycin A for all concentrations (*p* < 0.001). Compared to idebenone, compounds **5**, **8**, and **11** significantly increased MitoSOX fluorescence from 150 μM (*p* < 0.033), 75 μM (*p* < 0.033) and 75 μM (*p* < 0.002) onwards, respectively (Figure 3).

### 2.7. Colony Formation

Long-term toxicity was assessed using a standard colony formation assay, where the reference compound idebenone significantly reduced colony formation from 10 μM onwards (*p* < 0.002; Figure 4). Of the 11 test compounds, compounds **4** and **5** significantly reduced colony numbers from 20 μM (*p* < 0.001) and 10 μM (*p* < 0.002) onwards, respectively, while all other compounds already showed a significant reduction in colony numbers at 5 μM (*p* < 0.001; Appendix A). The effects of compounds **4** and **5** were significantly lower across all test concentrations (*p* < 0.001), compared to the other test compounds.

### 2.8. Nuclear Morphology

The previous results suggested that the observed toxicity at higher concentrations was mainly associated with a loss of cells and/or impaired cell membrane integrity, which is indicative of reduced proliferation and/or cell death. High content imaging was used to simultaneously quantify nuclei numbers per field of vision, nuclear size, and fluorescence intensity as markers of pyknosis, respectively. Using this analysis, the reference compound idebenone, like most test compounds (**1**–**3** and **8**–**10**), significantly reduced nuclei numbers (*p* < 0.001) and nuclear size (*p* < 0.033) and increased nuclear fluorescence (*p* < 0.002) at 100 µM (Figure 5). Although 4 compounds (**5**–**7** and **11**) significantly reduced nuclei numbers (**5**, *p* < 0.033; **6**, **7**, and **11**, *p* < 0.001), no significant changes to the nuclear size or fluorescence intensity were detected. Compound **4** did not significantly change nuclei number, size, or fluorescence intensity. Overall, the individual results of this approach appear mostly consistent in that those compounds which reduce nuclei numbers, also decrease nuclear area and increase nuclear fluorescence (Figure 5a–c).

### 2.9. DNA Damage

Due to the reported redox nature of the test compounds [29] and the observed reactive oxygen species (ROS) production by some compounds, the possibility of oxidative stress-induced DNA damage was assessed (Figure 6). For this purpose, the structurally related naphthoquinone menadione was used as a positive control [32]. In our test system, menadione significantly increased the number of γ-H_2_AX-positive cells from 20 μM onwards (*p* < 0.001), while no signs of DNA damage by idebenone were detected across all the concentrations tested (Figure 6). Of the 11 test compounds, **3** and **4** did not significantly increase the number of γ-H_2_AX-positive cells at any concentration, while 2 compounds increased the number of γ-H_2_AX-positive cells at 40 μM (**10**, *p* < 0.033; **11**, *p* < 0.001), 5 compounds increased γ-H_2_AX-positive cells from 30 μM onwards (**1**, **4**, and **9**, *p* < 0.002; **5** and **6**, *p* < 0.033), 1 compound showed an increase from 20 μM (**1**, *p* < 0.001), and 1 compound showed an increase from 10 μM onwards (**8**, *p* < 0.033). Increased numbers of γ-H_2_AX-positive cells by 6 compounds (**3**–**5** and **9**–**11**) were significantly lower compared to menadione across all test concentrations (*p* < 0.033) and no significant differences were observed between compounds **3**, **4**, and idebenone (Figure 6).

### 2.10. Transformation Potential

Based on the induction of γ-H_2_AX, indicative of DNA damage by some test compounds, their potential to transform substrate-dependent growth of HepG2 cells into substrate-independent cell growth by DNA mutations was assessed. To quantify transformation potential of our test compounds, a high throughput variant of the traditional semi-solid agar invasion assay was employed [33] using resorufin fluorescence as the indicator of cell growth [34]. The mutagenic compound 2-amino-1-methyl-6-phenylimidazo[4,5-b]pyridine (PhIP) was used as a positive control [35]. In our test system, PhIP exhibited its reported transformation potential by significantly increasing substrate-independent cell growth (*p* < 0.001, Figure 7), while neither idebenone nor any of the test compounds significantly increased cell growth at any test concentration. Similar to PhIP (*p* < 0.001), all reference and test compounds, except for compound **4**, significantly reduced resorufin fluorescence at 40 μM (*p* < 0.002) as a consequence of increased compound toxicity. Compared to idebenone, the reduction of resorufin fluorescence by compound **4** was significantly lower across all test concentrations (*p* < 0.002). At 40 μM, compounds **5** and **11** showed significantly lower inhibitory effects compared to idebenone, whereas all other compounds showed greater inhibition from 10–20 μM onwards (*p* < 0.002).

### 2.11. Summary of Results

Comparative in vitro toxicities of the test compounds **1**–**11** against the reference compound idebenone are summarized in Table 2.

## 3. Discussion

This study aimed to characterize the in vitro toxicity of the most promising compounds out of a novel range of cytoprotective and mito-protective short-chain quinones (SCQs) [29,30]. Due to the redox activity of the quinone moiety [29], this class of compounds is associated with an inherent risk of producing reactive oxygen species (ROS) [36,37], which could lead to toxicity and cell death at higher concentrations. In addition, the redox activity of quinones can generate false-positive results in many standard viability assays such as MTT and WST-1 [38,39]. Similarly, there is good evidence that redox reactions of SCQs can affect growth factor signalling [23,40] and cell proliferation, which can affect assays that rely on proliferation such as the colony formation assay. Consequently, the current study employed several distinct assays to assess different forms of toxicity including metabolic toxicity, loss of cell membrane integrity, mitochondrial ROS production, long-term toxicity, pyknosis, DNA damage, and transformation potential.

Based on data of related compounds in pre-clinical animal models and patients, the liver is expected to be exposed to the highest concentrations of SCQs [26,41]. In addition, some unrelated compounds have been reported to only show toxicity after metabolic conversion in the liver [42]. Therefore, the present study used a liver-derived cell line to account for this fact. Although HepG2 cells are less metabolically active compared to primary hepatocytes and other cell lines such as C3A or HepaRG, HepG2 cells are widely employed for in vitro toxicity studies due to their high phenotypic stability and unlimited availability for robust and reproducible outcomes [43,44]. It must be acknowledged that this approach cannot exclude tissue-specific toxicities such as neurotoxicity, so our data can only serve as a first approximation to compare the test compounds against reference compounds and against each other. Some test compounds have been successfully used in several animal models of different diseases, in both systemic (oral) and topical (eyedrops) applications over months, without any overt signs of toxicity (unpublished results). This could indicate that this chemical class is generally associated with low toxicity, comparable to the reference compound idebenone; however, this remains to be confirmed experimentally.

When metabolic toxicity was assessed, in general, ATP levels appeared a more sensitive readout compared to the WST-1 assay for all the compounds tested. This could indicate that the interaction of the redox active compounds interfered with the conversion of the WST-1 dye [39] or that the test compounds specifically affect ATP production. Due to this uncertainty, the WST-1 and similar assays should only be used with great caution when testing compounds with confirmed or suspected redox activity. Despite the metabolic limitation of HepG2 cells, our data clearly demonstrate the superior safety profile of the *L*-phenylalanine derivatives **4** and **5** compared to idebenone to enable us to progress these candidates in future studies. It is interesting to note that protein content per well, indicative of cell number, also appeared less sensitive than the measurement of ATP levels. This supports the idea that the test compounds at higher concentrations affect ATP levels in our test system while simultaneously leading to cell loss, presumably by a cell death pathway that involves pyknosis. The subsequent measurement of structural integrity of the cell membrane largely mirrored the toxicity observed with the ATP assay. Surprisingly, the commercial Multi-Tox Fluor assay displayed significantly lower sensitivity when measuring membrane integrity compared to a standard propidium iodide (PI) incorporation assay. The reason for this is not known. While we can only speculate that compared to simple diffusion and binding kinetics of a dye such as PI, the enzymatic conversion of a substrate underlies more restrictive conditions, which might be partially impaired under the specific conditions of our test system. It is interesting to note that the mitochondrial superoxide production did not correlate with the observed ATP data, which suggests that mitochondrial ROS is not responsible for the metabolic toxicity, while the observed acute toxicities were replicated by the colony formation assay that measures long-term toxicity. Overall, the results of the present study consistently demonstrate that only the two *L*-phenylalanine derivatives **4** and **5** show comparable or lower levels of toxicity compared to the reference compound idebenone across most endpoints utilized in this study.

Since most results in this study could be attributed to cell loss, we assessed structural changes in nuclear morphology (pyknosis), which is mostly indicative of apoptotic cell death [45]. Surprisingly, the results of this approach differed significantly from the results of the previous assays. In particular, the significantly lower induction of pyknosis by several test compounds compared to idebenone did not correlate with the previous toxicity assays (Table 2). For some compounds, the measured toxicity did not lead to a significant induction of pyknosis despite profound effects on protein levels, such as the tyramine derivatives **6** and **7** and the *L*-proline derivative **11**. The molecular reason for this obvious discrepancy is unclear at present and is subject to ongoing investigations but could either involve a different form of cell death such as necrosis or a difference in time course. When we assessed the DNA damaging potential of our test compounds, idebenone did not induce any DNA damage, while most test compounds showed a dose-dependent induction of DNA damage at higher concentrations. One exception was the *L*-phenylalanine derivative **4** that consistently showed low or absent toxicity throughout most assays. However, the structurally related *L*-phenylalaninol derivative **3** [30], also showed no significant induction of DNA damage, which was surprising given the consistent toxicity results from all other assays. Based on the observed DNA damaging activity of some test compounds, their transformation potential was assessed. The reference compound PhIP increased transformation only up to 30 μM in our test system, since cytotoxicity at higher concentrations has been reported [46]. In contrast, the clinically used idebenone [47,48] as well as the test compounds also did not appear to promote cellular transformation in this test system, but their toxicities were mirrored in the agar invasion assay in a concentration dependent manner.

Collectively, this study highlights the independence of the toxicity assays used and justifies a panel of assays to detect the different aspects of toxicity of a class of compounds during early drug development. The current study indicates that the carboxylic acid derivatives (i.e., **4** and **11**) are significantly less toxic than the corresponding alcohols (i.e., **3** and **10**, respectively). One possible explanation could be that oxidative metabolites of alcohols could show greater toxicity. However, the reported high metabolic stability of these compounds [30] seems to directly implicate the alcohol function in their increased toxicity in vitro. Although the current study does not allow any predictions towards toxicity in vivo, it is important to relate the observed results to expected in vivo concentrations. At present, achievable plasma or tissue levels for the test compounds are not known. However, for the chemically related reference compound idebenone, the highest achievable plasma concentrations (C_max_) in patients are in the single digit micromolar range [49]. While tissue levels in the central nervous system and retina are in the low nanomolar range [41,50], the highest concentrations were detected in the liver (~2 µM in rats; ~10 µM in dogs) [41]. However, these concentrations are only present for short periods of time (minutes to a few hours depending on the organism) due to the high rate of hepatic metabolism [26,50]. While our test compounds show significantly higher metabolic stability compared to idebenone [30], their structural similarity and solubility characteristics could indicate similar C_max_ values in vivo. For idebenone, detailed toxicity data is available from a large range of patients [18,26,27,51]. Despite the in vitro toxicity observed in the present study for concentrations above 5 µM (Table 2 and Appendix A), idebenone is extremely well tolerated up to concentrations of 2250 mg/patient/day with the most common adverse events described as reversible intestinal disturbances [26]. Hence, the observed toxicities for the test compounds in this study, even if higher than idebenone, cannot be interpreted as evidence for systemic toxicities at therapeutic doses. Nevertheless, the increased in vitro metabolic stability of our test compounds compared to idebenone could increase area under the curve (AUC) values [30] while altered chemical structures and solubilities could influence ADME characteristics. This highlights that the present study explored the concentrations from which toxicity can be expected and requires future pharmacokinetic studies of selected compounds in vivo to establish their safety margins.

Based on the current data and unpublished and ongoing studies, future studies will investigate the suitability of the most promising compounds to counteract mitochondrial dysfunction-induced pathologies.

## 4. Materials and Methods

### 4.1. Chemicals and Reagents

Idebenone was provided by Santhera Pharmaceuticals (Pratteln, Basel-Landschaft, Switzerland) as a reference compound. The novel naphthoquinone derivatives **1**–**11** were synthesized as described previously [29]. Dimethylsulfoxide (DMSO), Dulbecco Modified Eagle Medium (DMEM, D5523), sodium bicarbonate (NaHCO_3_), 2-[4-(2-hydroxyethyl)piperazin-1-yl]ethanesulfonic acid (HEPES), 1,4-dithiothreitol (DTT), magnesium chloride (MgCl_2_), bovine serum albumin (BSA), Triton X-100, paraformaldehyde (PFA), Tween-20, rat tail collagen, sodium hydroxide (NaOH), propidium iodide (PI), poly-L-lysine, menadione, 4′,6-diamidino-2-phenylindole (DAPI), noble agar, and resazurin sodium salt were purchased from Sigma-Aldrich (Ryde, NSW, Australia). Trypsin, ethylenediaminetetraacetic acid (EDTA), phosphate-buffered saline (PBS) tablets, Hanks Balanced Salt Solution (HBSS), MitoSOX Red, Hoechst 33342, Coomassie Brilliant Blue, and goat anti-mouse Alexa Fluor 488 secondary antibody (A-11029) were obtained from Thermo Fisher Scientific (Scoresby, VIC, Australia). Foetal bovine serum (FBS) was obtained from SAFC Biosciences (Brooklyn, VIC, Australia). WST-1 Assay Kit and 2-amino-1-methyl-6-phenylimidazo[4-b]pyridine (PhIP) were obtained from Cayman Chemical (Redfern, NSW, Australia). DC Protein Assay Kit was obtained from Bio-Rad Laboratories (Gladesville, NSW, Australia). *D*-luciferin, luciferase, and MultiTox-Fluor Multiplex Cytotoxicity Assay Kit were obtained from Promega (Alexandria, NSW, Australia). Mouse monoclonal anti-phospho-Histone H_2_AX (Ser139) antibody (JBW301) was obtained from Merck (Kilsyth, VIC, Australia). Methanol and acetic acid were obtained from VWR (Tingalpa, QLD, Australia). Cell culture plastics were obtained from Corning (Mulgrave, VIC, Australia), if not stated otherwise.

For all assays, stock solutions (100 mM in DMSO) of reference and test compounds were prepared as single use aliquots and stored at −20 °C until used. DMEM was prepared according to the manufacturer’s instructions, sterilized by filtration using 0.22 µm bottle top filters, supplemented with FBS (10%), NaHCO_3_ (3.7 g/L), and stored at 4 °C.

### 4.2. Cell Culture

Cryopreserved HepG2 cells (HB-8065, ATCC, Noble Park North, VIC, Australia) were passaged after thawing for at least 2–3 weeks to reach steady cumulative growth rates before being used for the experiments. The cells were routinely cultured in 5 mL DMEM (95% humidified air, 5% CO_2_, 37 °C) in cell culture flasks (25 cm^2^, 0.2 μm vent cap). The cells were passaged twice weekly when reaching ~75% confluency. Cell suspensions were generated by trypsinization (1 × wash with 5 mL PBS, 1 × 0.5 mL EDTA (0.5 mM, pH 8), 1 × 0.5 mL trypsin (0.25%, 3.5 min), and seeded into new T25 flasks at 8 × 10^4^ cells/cm^2^.

### 4.3. Multiplex Detection of NAD(P)H, ATP, and Protein Levels

The multiplex detection of NAD(P)H synthesis (absorption, 450 nm), ATP (luminescence) and protein levels (absorption, 750 nm) from individual wells was used to increase throughput and quality of results. No statistically significant differences were observed over a range of concentrations with or without NAD(P)H measurement prior to the quantitation of ATP and protein content from cell lysates (Appendix A). Briefly, 2 × 10^4^ cells were seeded in 100 μL DMEM per well in transparent 96-well plates and left to adhere overnight. Subsequently, cells were treated with test compounds for 24 h (0–200 μM in 25 μL DMEM). After incubation with 5 μL WST-1 reagent for 1 h, absorption was measured using a plate reader (Multiskan Go, Thermo Fisher Scientific, Scoresby, VIC, Australia). After media removal, cells were washed twice with 110 μL PBS and permeabilized for 10 min (0.5% Triton X-100/PBS, 40 μL) at room temperature. Cell lysates (10 μL) were mixed with reaction buffer (300 μM d-luciferin, 5 μg/mL luciferase, 25 mM HEPES, 75 μM DTT, 6.25 mM MgCl_2_, 625 μM EDTA, 1 mg/mL BSA in PBS, pH 7.4; 90 μL) in white 96-well plates, followed by immediate measurement of luminescence using a plate reader (Fluoroskan Ascent, Thermo Fisher Scientific, Scoresby, VIC, Australia). Lastly, protein contents from cell lysates (5 μL) were quantified using the DC Protein Assay as recommended by the manufacturer. Absorption and relative luminescence units (RLU) were standardized on the non-treated control cells (100%). A standard curve using BSA (0–2 mg/mL) was used for protein quantitation, and protein levels were standardized on the non-treated control (100%). Data was expressed as mean ± standard error of the mean (SEM) from independent experiments with 6 parallel wells per experiment.

### 4.4. Propidium Iodide Incorporation

Cell membrane integrity was assessed using the non-cell membrane permeable dye propidium iodide (PI). For this purpose, 1 × 10^4^ cells were seeded in 100 µL DMEM per well in 384-well plates (781091, µClear, Greiner, Ryde, NSW, Australia) and left to adhere overnight. Subsequently, the cells were treated with test compounds for 24 h (0–200 μM in 50 μL DMEM). After media removal, the cells were stained with PI solution for 30 min (5 μM in 50 μL PBS), before PI fluorescence (Ex/Em 545/600 nm) was quantified using a plate reader (Fluoroskan Ascent, Thermo Fisher Scientific, Scoresby, VIC, Australia). RFUs were standardized on the non-treated control and expressed as fold-induction. Data represented the mean ± SEM from 3 independent experiments with 4 parallel wells per experiment.

### 4.5. Multi-Tox Fluor Kit

Following exposure to test compounds, two different toxicity parameters were measured using the MultiTox-Fluor Kit according to the manufacturer’s instructions. For this purpose, 5 × 10^3^ cells were seeded in 100 µL DMEM per well in black 384-well plates (781091, µClear, Greiner, Ryde, NSW, Australia) and left to adhere overnight. Subsequently, the cells were treated with test compounds for 24 h (0–200 μM in 25 μL DMEM). After incubation with the protease substrate mix for 1 h (permeable GF-AFC and non-permeable bis-AAF-R110, 25 μL), fluorescence for GF-AFC (Ex/Em 400/505 nm) and bis-AAF-R110 (Ex/Em 485/520 nm) were measured, respectively, using a multimode plate reader (Tecan Spark 20M, Tecan, Port Melbourne, VIC, Australia). RFUs were standardized on the non-treated control (100%) and expressed as mean ± SEM from 3 independent experiments with 3 parallel wells per experiment.

### 4.6. MitoSOX

To measure mitochondrial superoxide production, 384-well plates (781091, µClear, Greiner, Ryde, NSW, Australia) were coated with poly-*L*-lysine for 45 min (1:20 in HBSS, pH 7.4, 50 μL/well) before 9 × 10^3^ cells were seeded in 50 µL DMEM per well, left to adhere for 3 h, and loaded with MitoSOX Red (1 μM) and Hoechst 33342 (2 μg/mL in HBSS + 1% BSA, 30 μL/well) for 30 min (Appendix A). After treatment with test compounds for 30 min (0–200 μM in HBSS + 1% BSA, 50 μL/well), fluorescence (Ex/Em 355/600 nm) was quantified using a plate reader (Fluoroskan Ascent, Thermo Fisher Scientific, Scoresby, VIC, Australia). Antimycin A was used as a positive control [31]. RFUs were standardized on the non-treated control and expressed as fold-induction. Data represented the mean ± SEM from 3 independent experiments with 8 eight parallel wells per experiment.

### 4.7. Colony Formation Assay

2.5 × 10^3^ cells were seeded in 2 mL DMEM per well in 6-well plates and left to adhere overnight. Subsequently, cells were treated with test compounds for 14 days (0–100 μM). After media removal, the colonies were fixed for 10 min (4% PFA/PBS, 2 mL/well), stained for 10 min (1% Coomassie Brilliant Blue in 50% methanol and 10% acetic acid, 2 mL/well), before the colonies (>50 cells) were counted under a light microscope. Colony numbers were standardized on the non-treated control (100%) and expressed as mean ± SEM from 3 independent experiments with 4 four parallel wells per experiment.

### 4.8. Assessment of Changes in Nuclear Morphology

To quantitate nuclear morphology, 384-well plates (781091, µClear, Greiner, Ryde, NSW, Australia) were coated with rat tail collagen for 45 min (1:20 in HBSS, pH 7.4, 50 μL/well) before 1 × 10^4^ cells were seeded in 100 µL DMEM per well and left to adhere overnight. After treatment with test compounds for 24 h (100 μM in 50 μL HBSS), fixation for 10 min (4% PFA/PBS, 50 μL/well) and permeabilization for 10 min (0.5% Triton X-100/PBS, 50 μL/well), the cells were stained with DAPI for 2 min (1:10,000 in 0.1% Tween-20/PBS, 15 μL/well). After washing three times for 5 min (PBST, 50 μL/well), cells were stored in PBS (50 μL) for high content imaging using an IN Cell 2200 analyser (10 × magnification, GE Healthcare, Rydalmere, NSW, Australia). Morphological changes (area and intensity) were automatically quantified for each individual nucleus. Images acquired from 4 wells with 2 images each were automatically analysed using IN Carta image analysis software (GE Healthcare, Rydalmere, NSW, Australia). Nuclear intensity was standardized on the average intensity of non-treated control nuclei and expressed as fold-change. Data represented the mean ± SEM of quadruplicates and ~1000 cells were analysed per treatment.

### 4.9. Assessment of DNA Damage

To assess DNA damage, 5 × 10^3^ cells were seeded in 100 µL serum-free DMEM per well in 384-well plates (781091, µClear, Greiner, Ryde, NSW, Australia) pre-coated with rat tail collagen as described above and left to adhere overnight. The cells were treated with test compounds for 4 h (0–40 μM in 100 μL HBSS/well) while menadione was used as a positive control [32]. After fixation (50 μL) and permeabilization (50 μL) as described above, unspecific antibody binding was blocked for 1 h (5% FBS + 5% BSA in PBS, 50 μL/well) before the samples were exposed to mouse monoclonal anti-phospho-Histone H_2_AX (Ser139) antibody overnight (1:1000 in blocking buffer, 15 μL/well). After exposure to goat anti-mouse Alexa Fluor 488 secondary antibody for 1 h (1:10,000 in PBST, 15 μL/well), nuclei were counterstained using DAPI and stored for imaging and analysis as described above. The average numbers of γ-H_2_AX-positive cells were automatically quantified for all acquired images using IN Carta image analysis software (GE Healthcare, Rydalmere, NSW, Australia). Results were standardized on the non-treated control and expressed as fold-change. Data represented the mean ± SEM of quadruplicate images from 3 independent assays. At least 1000 cells were analysed per treatment.

### 4.10. Agar Invasion Assay

To further investigate if the test compounds can induce mutations in cells at previously tested concentrations (10–40 μM), the agar invasion assay in 384-well format (781091, µClear, Greiner, Ryde, NSW, Australia) was performed as previously described [33]. PhIP was used as a positive control [35]. Cell numbers and incubation times were optimized for PhIP and used for all the test compounds. Two hundred cells were seeded in 50 µL DMEM (0.4% agar supplemented) per well in 384-well plates pre-coated with solidified agar (0.6%, 10 µL). The plates were left to solidify for 1 h at room temperature and incubated overnight at 37°C before cells were treated for 21 days with reference and test compounds (10–40 μM in 15 µL DMEM/well). Subsequently, the cells were stained with resazurin for 4 h (440 µM in 7 µL PBS/well) [34] before fluorescence (Ex/Em 545/600 nm) was quantified using a plate reader (Fluoroskan Ascent, Thermo Fisher Scientific, Scoresby, VIC, Australia). RFUs were standardized on the non-treated control and expressed as fold-induction. Data represented the mean ± SEM from 3 independent experiments with 3 parallel wells per experiment.

### 4.11. Statistical Analysis

Using GraphPad Prism (version 8.2.1, San Diego, CA, USA), one- or two-way ANOVA followed by Dunnett’s multiple comparison post-test was performed to compare between compounds or concentrations: *** *p* < 0.001, ** *p* < 0.002, * *p* < 0.033, otherwise non-significant; non-linear regressions were generated and half maximal inhibitory concentrations (IC_50_) were automatically calculated by the software.

## 5. Conclusions

This study characterized the in vitro toxicity of the most promising cytoprotective and mito-protective short-chain naphthoquinones [29,30]. The multiplex detection of compatible assays described in this study provides a convenient, cost-effective, and rapid approach to increase throughput. Overall, the test compounds, with some exceptions, showed largely comparable results between different assays. However, standard assays/dyes appeared to be associated with significantly higher sensitivity compared to commercially available kits. Compared to the other test compounds, the *L*-phenylalanine derivative **4** showed the most promising safety profile, with lower metabolic toxicity, lower effects on membrane integrity, lower long-term toxicity, as well as an absence of mitochondrial toxicity, pyknosis, DNA damage, or transformation potential. Our results highlight the importance of using a set of independent assays to assess distinct toxicity profiles to characterize a class of compounds. Importantly, this study increased our understanding of the comparative toxicities of the range of test compounds and supports the development of the most promising short-chain naphthoquinone(s) towards their clinical use.

## Figures and Tables

**Figure 1 pharmaceuticals-13-00184-f001:**
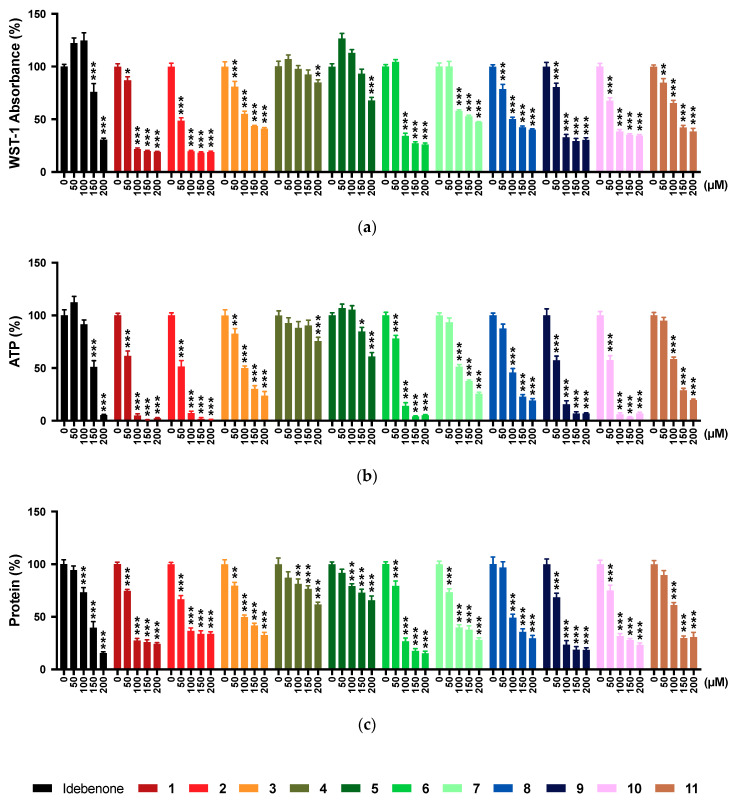
Effect of test compounds on metabolic toxicity. Cells were exposed to reference and test compounds (0–200 µM) for 24 h before (**a**) WST-1 absorption; (**b**) ATP levels; (**c**) protein contents were quantified. Data represents mean ± SEM of 3 independent experiments with 6 parallel wells per experiment. Two-way ANOVA was performed to compare test concentrations against the non-treated control: *** *p* < 0.001, ** *p* < 0.002, * *p* < 0.033. Full datasets shown in Appendix A.

**Figure 2 pharmaceuticals-13-00184-f002:**
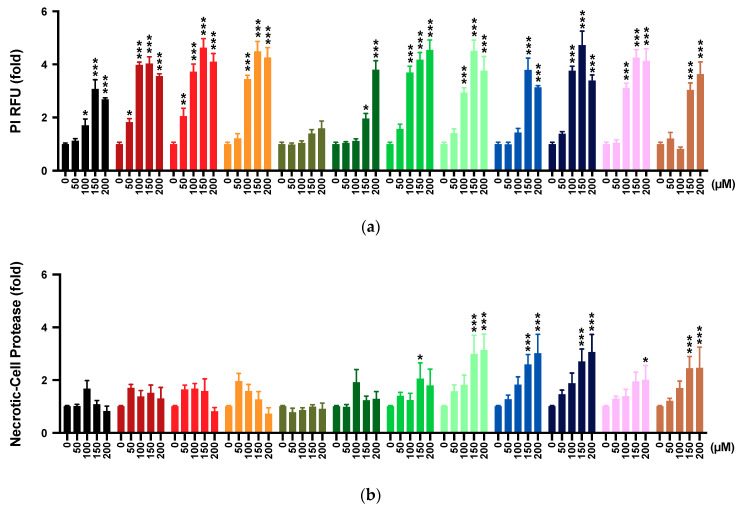
Effect of test compounds on membrane integrity. Cells were exposed to reference or test compounds (0–200 µM) for 24 h before (**a**) propidium iodide (PI) incorporation; (**b**) necrotic-cell protease activity, and (**c**) viable-cell protease activity (Multi-Tox Fluor Kit) were assessed. Data represents mean ± SEM from 3 independent experiments with 4 parallel wells per experiment. Two-way ANOVA was performed to compare test concentrations against the non-treated control: *** *p* < 0.001, ** *p* < 0.002, * *p* < 0.033. Full datasets shown in Appendix A. RFU, relative fluorescence units.

**Figure 3 pharmaceuticals-13-00184-f003:**
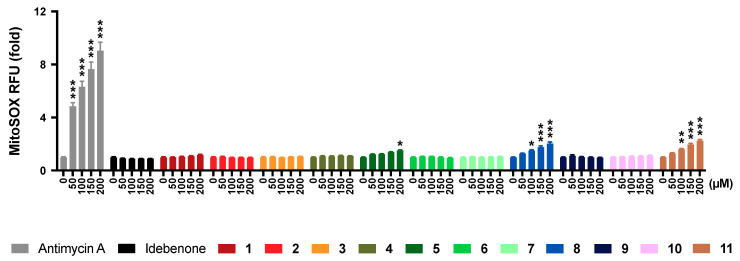
Effect of test compounds on mitochondrial superoxide production. Cells were exposed to reference (antimycin A, idebenone) or test compounds (0–200 µM) for 30 min before mitochondrial superoxide levels were quantified. Data was expressed as mean ± SEM of 3 independent experiments with 8 parallel wells per experiment. Two-way ANOVA was performed to compare test concentrations against the non-treated control: *** *p* < 0.001, ** *p* < 0.002, * *p* < 0.033. Full datasets available in Appendix A.

**Figure 4 pharmaceuticals-13-00184-f004:**
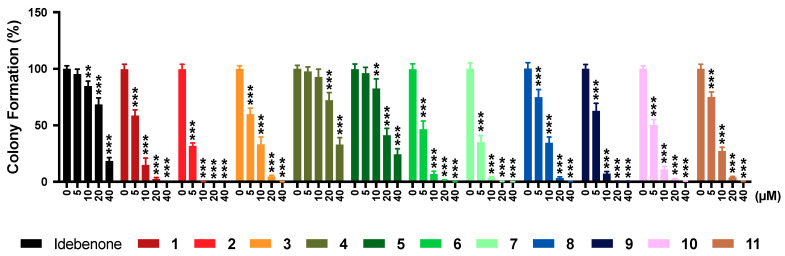
Effect of test compounds on colony formation. Cells were exposed to reference or test compounds (0–100 µM) for 14 days before colonies (>50 cells) were quantified. Data was expressed as mean ± SEM of 3 independent experiments with 4 parallel wells per experiment. Two-way ANOVA was performed to compare test concentrations against the non-treated control: *** *p* < 0.001, ** *p* < 0.002. Full datasets available in Appendix A.

**Figure 5 pharmaceuticals-13-00184-f005:**
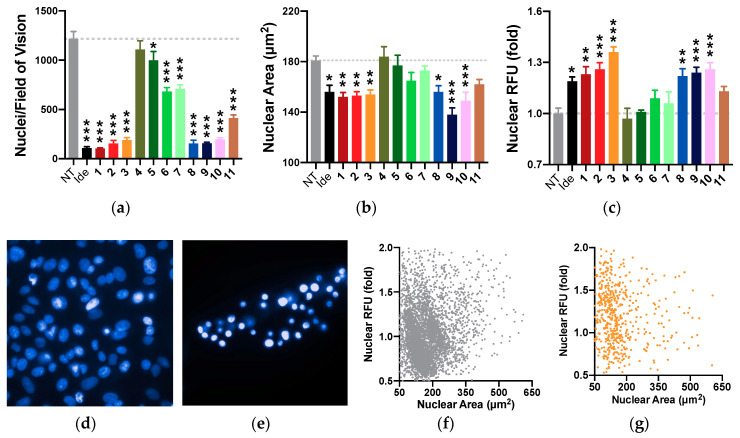
Effect of test compounds on nuclear count and morphology. Cells were exposed to reference or test compounds (100 µM) for 24 h before (**a**) nuclear count, (**b**) area, and (**c**) intensity were assessed. Exemplary fluorescence images (60 × magnification) of (**d**) non-treated (NT) and (**e**) treated DAPI-stained nuclei; single cell plots of nuclei either (**f**) NT or (**g**) treated with compound **3** are presented. Compound **3** (**e**,**g**) significantly reduced average nuclear count and size, and increased average fluorescence intensity than NT (**d**,**f**). Nuclear RFU was standardized on the average intensity of NT control nuclei and expressed as fold-change. Data represents mean ± SEM of 8 independent images per treatment. Two-way ANOVA was performed to compare idebenone or test compounds against the non-treated control: *** *p* < 0.001, ** *p* < 0.002, * *p* < 0.033.

**Figure 6 pharmaceuticals-13-00184-f006:**
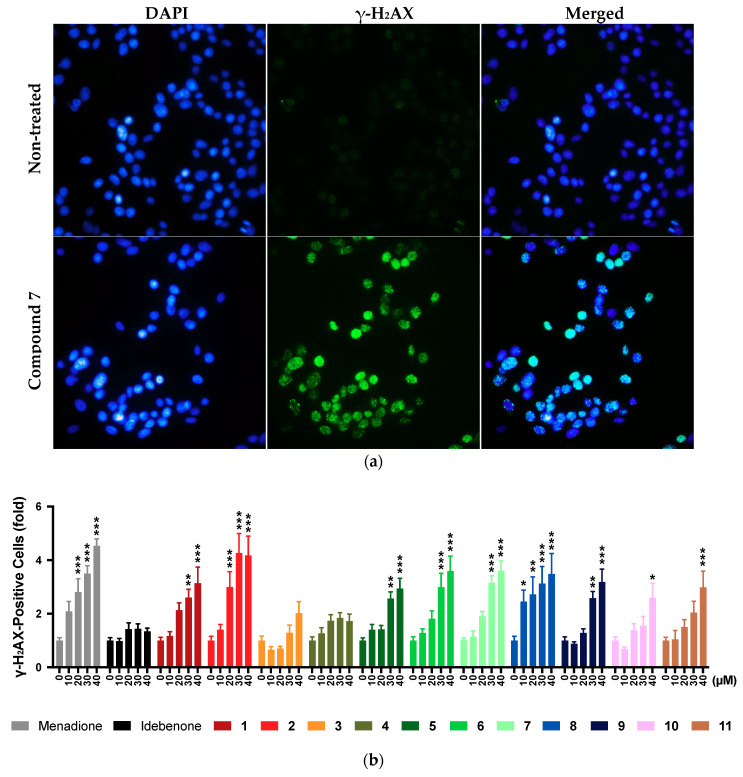
Effect of test compounds on DNA damage. Cells were exposed to reference compounds (menadione, idebenone) or test compounds (0–40 µM) for 4 h before the presence of nuclear γ-H_2_AX positive cells was quantified. (**a**) Exemplary images (60 × magnification) used for quantitation of γ-H_2_AX-positive cells using compound **7** as positive treatment and (**b**) quantitation of results for all reference and test compounds. Data represent the mean ± SEM of 3 independent experiments with 4 parallel wells per experiment. Overall, >1000 cells were analysed per treatment. Two-way ANOVA was performed to compare test concentrations against the non-treated control: *** *p* < 0.001, ** *p* < 0.002, * *p* < 0.033.

**Figure 7 pharmaceuticals-13-00184-f007:**
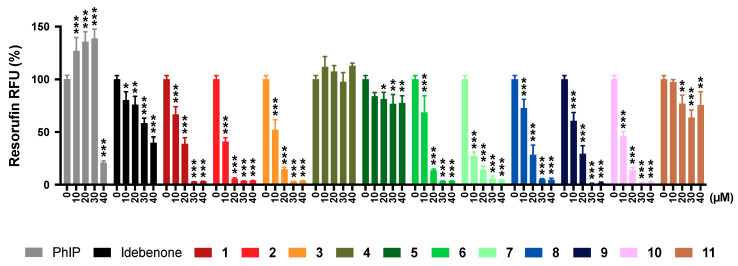
Transformation potential of test compounds. Cells were exposed to reference compounds (PhIP, idebenone) or test compounds (0–40 µM) for 21 days in soft agar before cell growth under these conditions was quantified. Data represents the mean ± SEM of 8 independent wells per treatment. Two-way ANOVA was performed to compare test concentrations against the non-treated control: *** *p* < 0.001, ** *p* < 0.002, * *p* < 0.033.

**Table 1 pharmaceuticals-13-00184-t001:** Chemical structure, physical properties, in vitro efficacy, and stability of the benzoquinone idebenone and 11 novel naphthoquinone derivatives.

Compound	Structure	N	R	Formula	Molecular Weight (g/mol)	LogP ^1^	LogD ^2^	In Vitro Cytoprotection ^3^	In Vitro Metabolic Stability ^4^
%	*p*-Value	%	*p*-Value
Idebenone	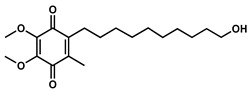	-	-	C_19_H_30_O_5_	338.4	1.24	3.57	66.2 ± 12.0	-	27.3 ± 3.9	-
**1**(UTAS#81)	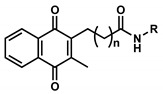	2	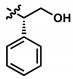	C_23_H_23_NO_4_	377.4	2.24	2.81	83.8 ± 19.9	0.191	92.6 ± 16.9	<0.001
**2**(UTAS#80)	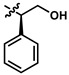	C_23_H_23_NO_4_	377.4	2.24	2.81	87.6 ± 19.7	0.025	96.6 ± 11.1	<0.001
**3**(UTAS#62)	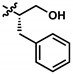	C_24_H_25_NO_4_	391.5	2.52	3.10	93.1 ± 13.7	<0.001	84.0 ± 15.5	<0.001
**4**(UTAS#37)	2	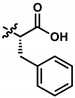	C_24_H_23_NO_5_	405.4	2.48	0.12	100.3 ± 17.3	<0.001	96.0 ± 7.5	<0.001
**5**(UTAS#72)	3	C_25_H_25_NO_5_	419.5	2.90	0.74	90.7 ± 15.6	0.146	91.4 ± 0.8	<0.001
**6**(UTAS#74)	2	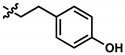	C_23_H_23_NO_4_	377.4	2.67	3.43	91.7 ± 15.6	0.101	45.7 ± 2.9	0.034
**7**(UTAS#88)	3	C_24_H_25_NO_4_	391.5	3.09	3.87	91.8 ± 9.8	0.097	60.3 ± 1.7	<0.001
**8**(UTAS#54)	2	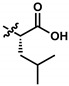	C_21_H_25_NO_5_	371.4	2.04	0.26	98.7 ± 10.9	0.004	84.3 ± 9.2	<0.001
**9**(UTAS#77)	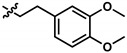	C_25_H_27_NO_5_	421.5	2.80	3.41	95.9 ± 19.4	0.017	58.3 ± 11.0	<0.001
**10**(UTAS#61)	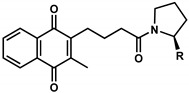	-	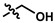	C_20_H_23_NO_4_	341.4	1.06	1.71	100.7 ± 28.4	0.002	78.9 ± 7.4	<0.001
**11**(UTAS#43)	-	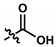	C_20_H_21_NO_5_	355.4	1.02	-1.32	92.7 ± 7.6	0.018	67.2 ± 2.3	<0.001

^1^ LogP was predicted using ChemDraw Professional software (version 16.0, PerkinElmer, MA, USA). ^2^ LogD was predicted using MarvinView software (version 19.25, ChemAxon, Budapest, Hungary). ^3^ In vitro cytoprotection of HepG2 by 10 μM SCQs against rotenone-induced mitochondrial complex I dysfunction. Cytoprotection was calculated as a relative percentage of cell survival compared to untreated cells (26.9 ± 7.9%). Data was expressed as mean ± standard deviation (SD) [29]. ^4^ In vitro metabolic stability of 40 µM SCQs over 6 h on HepG2. Stability was calculated as percentage compounds found remaining after 6 h. Data was expressed as mean ± SD [30].

**Table 2 pharmaceuticals-13-00184-t002:** Summary of the in vitro toxicity of compounds.

Compound	Multiplex Detection			Membrane Integrity ^2^	Multi-Tox Fluor Assay		Mitochondrial Superoxide ^2^	Colony Formation ^1^	Pyknosis	DNA Damage ^2^	Transformation Potential
	WST-1 ^1^	ATP ^1^	Protein ^1^		Necrotic-Cell Protease ^2^	Viable-Cell Protease ^1^					
Idebenone	151.7 ± 5.9	146.8 ± 14.2	136.8 ± 4.7	≥100	N	71.1 ± 11.1	N	26.0 ± 5.2	Y	N	N
**1**	59.0 ± 2.2	59.0 ± 4.7	59.6 ± 0.6	≥50	N	55.8 ± 9.1	N	6.2 ± 1.8	Y	≥30	N
**2**	45.7 ± 1.3	45.5 ± 3.8	52.7 ± 7.1	≥50	N	50.7 ± 7.1	N	4.8 ± 0.1	Y	≥20	N
**3**	88.8 ± 8.8	95.4 ± 9.2	66.1 ± 7.2	≥75	N	54.9 ± 7.6	N	7.1 ± 3.2	Y	N	N
**4**	>200	>200	>200	N	N	>200	N	31.2 ± 10.5	N	N	N
**5**	>200	>200	>200	≥150	N	161.5 ± 15.2	≥200	20.6 ± 5.9	N	≥30	N
**6**	69.6 ± 1.8	67.0 ± 4.7	66.3 ± 4.9	≥75	≥125	52.0 ± 9.3	N	4.7 ± 1.1	N	≥30	N
**7**	78.0 ± 5.6	78.2 ± 4.4	60.0 ± 8.3	≥100	≥125	56.1 ± 8.2	N	4.2 ± 1.0	N	≥30	N
**8**	83.1± 3.7	88.8 ± 10.5	80.8 ± 3.3	≥125	≥125	155.8 ± 15.7	≥100	8.1 ± 2.3	Y	≥10	N
**9**	55.4 ± 6.7	52.3 ± 4.8	57.3 ± 9.2	≥75	≥125	49.0 ± 11.8	N	5.6 ± 1.1	Y	≥30	N
**10**	51.5 ± 9.6	55.9 ± 12.0	61.2 ± 7.7	≥75	≥200	57.9 ± 7.8	N	4.8 ± 0.5	Y	≥40	N
**11**	99.7 ± 5.8	108.0 ± 19.2	91.8 ± 5.1	≥150	≥125	>200	≥100	7.4 ± 1.4	N	≥40	N

^1^ Data represents half maximal inhibitory concentrations (IC_50_) ± SD (μM) calculated using GraphPad Prism (version 8.2.1, San Diego, CA, USA). ^2^ Data represents the lowest observable concentration (μM) to cause statistically significant effects. Full datasets shown in Appendix A. Y, detected; N, not detected.

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
