# Peer review of "Comparative In Vitro Toxicology of Novel Cytoprotective Short-Chain Naphthoquinones"

_pharmaceuticals, 2020, doi:10.3390/ph13080184_

Round 1
Reviewer 1 Report
Authors' response to Reviewer 1.
Point 1: “In the Introduction Section: “Leigh syndrome” was repeated twice.”
Response 1: We agree with the reviewer and have corrected this error. The repeated words have been deleted (Line 30).
Point 2: “Mitochondrial disorders are rare. In the Introduction Section, it says “large number of patients that …”. Please provide references about the incidence and prevalence of patients with mitochondrial disorders.”
Response 2: Thank you very much for this comment. Mitochondrial diseases are indeed rare, however “large number of patients” here referred to the admittedly rare mitochondrial diseases AND additionally also to numerous very common neurodegenerative conditions (such as Alzheimer’s disease, Parkinson’s disease, and multiple sclerosis), inflammatory conditions (such as ulcerative colitis), metabolic conditions (such as obesity and diabetes) as well as pretty much all ophthalmological conditions (such as glaucoma, diabetic retinopathy, age-related macular degeneration, and cataracts). Taken together, this represents a very large number of potential patients characterized by mitochondrial dysfunction that could potentially benefit from mito-protective treatments. We observed striking efficacy of two of our test compounds in a model of diabetic retinopathy and also in models of ulcerative colitis (manuscripts ready to be submitted), which support this point (since these studies are not published yet, we were unable to refer to them in the manuscript). The “large number of patients” has now been clarified in the revised version of the manuscript (Lines 33-39).
Point 3: “Please provide information about the results of Phase II studies of vatiquinone for FA and LS.”
Response 3: We appreciate this comment and agree with the reviewer. At present, evidence for therapeutic effects of vatiquinone on FA and LS are limited. The FA trial (NCT01962363) reported improved neurological function by vatiquinone (3 × 400 mg for 18 months) (Sullivan et al 2016). In the LS trial, the originally referenced clinical trial (NCT01962363) was estimated to complete by December 2021, thus in the manuscript another completed trial (NCT01721733) which was added for consistency. This study reported improved movement disorder by vatiquinone (3 × 100 mg for 6 months) (Martinelli et al 2012). Although, these two studies reported some therapeutic efficacy, Sullivan et al (2016) was not placebo-controlled and only compared therapeutic efficacy against historical records. In addition, Martinelli et al (2012) only described a single arm open-label study. Although, this is a very vague evidence base, the results of these two Phase II studies have now been included in the revised version of the manuscript (Lines 44-45) as suggested by the reviewer. In addition, the clinical results with the related compound sonlicromanol have also been added to the revised version of the manuscript (Lines 47-49), which added 4 new references. In MELAS and MIDD patients (NCT02909400), sonlicromanol (2 × 100 mg for 28 days) was reported to be well tolerated and safe (Janssen et al 2019). Likewise, sonlicromanol (800 mg for 7 days) was also reported to be well tolerated in LS and LHON patients (NCT02544217) (Koene et al 2017). Although, both studies reported favorable safety profiles, Janssen et al were unable to observe significant improvements as no formal primary end points were defined prior to this exploratory study, while Koene et al did not provide any efficacy data. Thus, at present the therapeutic profile of sonlicromanol is unclear.
Point 4: “What is the clear/scientific definition of mito-protection or mito-protective activity? I think these phrases are used in a lay-person language versus a scientific term.”
Response 4: We appreciate the reviewer’s comment. With increasing attention on mitochondrial disorders and mitochondrial functional impairment in a variety of diseases, the two terms “mito-protection” and “mito-protective activity” are increasingly used in different contexts in the scientific literature since their first academic use in 2013 (Brooks et al 2013; Prakash et al 2013). To clarify this point, we defined “mito-protection” and “mito-protective activity” in the revised version of the manuscript as “protection of cell viability against mitochondrial dysfunction” (Lines 56-57). Brooks, M.M.; Neelam, S.; Fudala, R.; Gryczynski, I.; Cammarata, P.R. Lenticular mitoprotection. Part A: Monitoring mitochondrial depolarization with JC-1 and artifactual fluorescence by the glycogen synthase kinase-3beta inhibitor, SB216763. Mol Vis 2013, 19, 1406-1412. Prakash, A.; Kumar, A. Mitoprotective effect of Centella asiatica against aluminum-induced neurotoxicity in rats: possible relevance to its anti-oxidant and anti-apoptosis mechanism. Neurol Sci 2013, 34, 1403- 1409, doi:10.1007/s10072-012-1252-1.
Point 5: “How pharmacologically relevant are the concentrations of these drugs at 50-150 micromolar? I am concerned about drugability of these agents. For example, vitamin K (menadione) that is commonly used in clinical practice, according to the authors can cause DNA damage at concentrations > 30 micromolar. Due to RedOx activity of naphthoquinone, these high micromolar concentration will most likely have substantial negative effect on cardiac and skeletal muscles.”
Response 5: Thank you very much for bringing up this important point. We fully agree with the reviewer that the concentrations used in this study are very high when compared to the reported plasma and tissue levels for SCQs. For example, for the reference compound idebenone plasma levels of only 1480 ng/mL in adults and tissue levels in the nM range have been reported (Becker et al 2010). It is important to state that this study aimed to work towards understanding the safety margins of these compounds that included the calculation of in vitro IC50 values (Table 2). For this purpose, test compound concentrations were increased until toxicity could be detected (or the max concentration was reached). We fully agree that the concentrations used in this study were far above concentrations that will ever be achieved in vivo. A point in case is the reference compound idebenone that showed “toxicity” in our test systems despite an extremely benign side effect profile in patients for doses of up to 2.25 g/day (Becker et al 2010). Only in combination with future PK studies for selected compounds can the data be interpreted towards in vivo toxicity. This point has now been discussed in the revised version of the manuscript (Lines 65-68, 367-387). Becker, C.; Bray-French, K.; Drewe, J. Pharmacokinetic evaluation of idebenone. Expert Opin Drug Metab Toxicol 2010, 6, 1437-1444, doi:10.1517/17425255.2010.530656
Point 6: “Figure 4 needs clarification about the interpretation of the 4d-4g.”
Response 6: We agree with the referee and have reworded the interpretation of Figure 5d-5g (originally Figure 4d-4g) for clarity in the text (Lines 205-217) and legend (Lines 225-229).
Point 7: “Please provide clearer explanation on the “Transformation” experiment and interpretation of the data.”
Response 7: The revised version of the manuscript now contains new texts in the Results (Lines 261-269, 271, 273, 276), Discussion (Lines 356-357), Materials and Methods (Lines 512-513) sections that explain the cell transformation assay and data interpretation more clearly.
Point 8: “The rationale behind choosing a liver-derived cells is flawed. There is a significant and incomparable difference between a complex organ such as liver and a cell line in vitro.”
Response 8: Thank you very much for this comment. While we agree with the referee that a cell line in vitro can never recapitulate the function of an entire organ in vivo, we disagree with the referee about the experimental approach we have taken. Hepatic cancer cell lines unmodified or genetically altered are in fact very commonly used as screening tools to determine comparative toxicity of test compounds. Especially HepG2 cells are widely used as cellular reference model for pharmaceutical studies to develop new drugs and understand drug metabolism. While the choice of cell line can be argued about (Qiu et al 2015), HepG2 cells in particular, express the majority of drug-metabolising enzymes (Knasmuller et al 1998; Castell et al 2006). Given that many drugs only gain toxicity after metabolic conversion, this activity is essential for drug toxicity testing. This study can certainly not replace in vivo toxicity testing and does not aim to do so. The aim of this study was a comparison of drug candidates against the well tolerated reference compound idebenone to identify possible development candidate molecules or highlight toxicities associated with specific chemistries. As such, this study will be followed by in vivo assessment of toxicity in animal models. This point has been clearly mentioned in the manuscript (Lines 305-311).
Qiu, G.H.; Xie, X.; Xu, F.; Shi, X.; Wang, Y.; Deng, L. Distinctive pharmacological differences between liver cancer cell lines HepG2 and Hep3B. Cytotechnology 2015, 67, 1-12, doi:10.1007/s10616-014-9761-9.
Knasmüller, S.; Parzefall, W.; Sanyal, R.; Ecker, S.; Schwab, C.; Uhl, M.; Mersch-Sundermann, V.; Williamson, G.; Hietsch, G.; Langer, T., et al. Use of metabolically competent human hepatoma cells for the detection of mutagens and antimutagens. Mutat Res 1998, 402, 185-202, doi:10.1016/s0027-5107(97)00297-2.
Castell, J.V.; Jover, R.; Martínez-Jiménez, C.P.; Gómez-Lechón, M.J. Hepatocyte cell lines: their use, scope and limitations in drug metabolism studies. Expert Opin Drug Metab Toxicol 2006, 2, 183-212, doi:10.1517/17425255.2.2.183.
Point 9: “Discussion is very long. I would focus more on the structure-activity relationship and potential mechanisms behind the observed data.”
Response 9: We agree and thank the referee for this comment. We have addressed the observed structure-activity relationship in the revised version of the manuscript (Lines 363-367) and reduced the discussion length. However, the inclusion of the points the referee raised above into the discussion required additional text sections. In total, the overall length of the discussion was reduced by 13 lines.
-------------------------------------------------------------------------------
Reviewer 1' comments to authors' response.
I reviewed the authors' responses and found all of them appropriate and adequate. I reviewed the changes in the revised typescript; they are all acceptable.
Reviewer 2 Report
Authors' response to Reviewer 2.
Point 1: “The authors present a compelling array of toxicity assays for their 11 experimental compounds. There are only two limitations to the study that should be noted by the authors. First, HepG2 cells are not as metabolically active as HepG2 C3A or even primary hepatocytes, so there is some limitation in the metabolic assay output. Second, the authors are targeting mitochondrial function and cytoprotection, yet not assessment of mitochondria is made by these assay. The addition of TMRE, mitoSOX, or even mitochondrial morphology MitoTracker or mito health markers such as pink, parkin, or mtDNA damage would enhance this analysis.”
Response 1: Thank you very much for pointing this out. In terms of enzymatic activity, HepG2 cells are indeed not comparable to C3A cells and primary hepatocytes. This limitation has been addressed in the revised version of the manuscript (Lines 305-311, 321-323). Though HepG2 cells are less metabolically active, this cell line is widely employed for in vitro toxicity studies due to their high phenotypic stability and unlimited availability. In addition, HepG2 express the majority of liver enzymes, making this cell line perfectly suited to provide a robust and reproducible test platform.
We also agree with your suggestions to include assays targeting mitochondrial function and have therefore included the results of a MitoSOX assay in the revised manuscript that assessed mitochondrial superoxide production. This assay is described at multiple places in the revised version of the manuscript (Lines 170-188, 286, 333-336, 465-474) and supplementary materials (Figure S2, Table S7). Our new data suggest that only a few test compounds elevate mitochondrial superoxide levels at high concentrations and support the conclusion that the L-phenylalanine derivative 4 demonstrated the lowest toxicity across all assays.
Point 2: “Two other general issues to consider. The physical growth characteristics of HepG2 cells make can have them grow into towers or “mini-organs” that create 3D structures that are harder to penetrate with dyes and drugs. Did the cells grow in this manner, and how would this more 3D like growth impact the penetrance of their drugs and their dye markers like the Multi-Tox. Finally, quinone structures can also act as endocrine disrupting agents, so future work may want to consider ER, AR, etc activity to minimize clinical side effects.”
Response 2: Thank you very much for these comments. Indeed, the nature that HepG2 cells like to grow in 3D clusters also drew our attention in the design and validation of our assays. During these processes, we initially tested a wide range of cell densities and chose the most appropriate in 6-, 96-, and 384-well formats for each assay (data not shown). For example, as depicted in Figure 4d, 4e and 5a, the confluency of cells was strictly controlled throughout our assays to make sure no 3D clusters were formed (see Materials and Methods section for detail). In particular, the use of high content imaging confirmed the suitability of the growth pattern of the cells for our analysis (if they would have grown in 3D patterns, it would have been impossible to do automated image analysis). To address this potential source of artefacts, we have added a new Figure S2 to the supplementary materials that displays the uniform growth pattern, which was also achieved by coating the plates with either collagen or lysine. Especially, the coating induced a more spread out phenotype with no 3D growth at all. As a consequence, this did not impact the credibility of our results including those from the Multi-Tox assay. Also, we appreciate your future directions very much. In fact, in one of our in vivo models, we have shown that one of our compounds potently protects against ER stress. We will certainly take up your comments and also include additional toxicity endpoints (including the possible endocrine disrupting activity) which will be beneficial to the progress our selected candidates.
Point 3: “Minor issues: Leigh syndrome listed twice in introduction”
Response 3: Thank you very much for pointing out this error. The repeated words in the introduction have been deleted (Line 30).
---------------------------------------------------------------------
Reviewer 2' comments to authors' response.
The changes enhance the article and make it appropriate for publication.
This manuscript is a resubmission of an earlier submission. The following is a list of the peer review reports and author responses from that submission.
Round 1
Reviewer 1 Report
This article compares 11 naphthoquinones with Idebenone in a series of experiments aiming at understanding the toxicity profile of these NQ.
Please see below:
1) In the Introduction Section: “Leigh syndrome” was repeated twice.
2) Mitochondrial disorders are rare. In the Introduction Section, it says “large number of patients that …”. Please provide references about the incidence and prevalence of patients with mitochondrial disorders.
3) Please provide information about the results of Phase II studies of vatiquinone for FA and LS.
4) What is the clear/scientific definition of mito-protection or mito-protective activity? I think these phrases are used in a lay-person language versus a scientific term.
5) How pharmacologically relevant are the concentrations of these drugs at 50-150 micromolar? I am concerned about drugability of these agents. For example, vitamin K (menadione) that is commonly used in clinical practice, according to the authors can cause DNA damage at concentrations > 30 micromolar. Due to RedOx activity of naphthoquinone, these high micromolar concentration will most likely have substantial negative effect on cardiac and skeletal muscles.
6) Figure 4 needs clarification about the interpretation of the 4d-4g.
7) Please provide clearer explanation on the “Transformation” experiment and interpretation of the data.
8) The rationale behind choosing a liver-derived cells is flawed. There is a significant and incomparable difference between a complex organ such as liver and a cell line in vitro.
9) Discussion is very long. I would focus more on the structure-activity relationship and potential mechanisms behind the observed data.
Reviewer 2 Report
This study is devoted to the investigation of the so-called toxicity of a set of naphthoquinones, previously synthesized by the authors (refs 24 and 25), using a plethora of different assays ranging from WST-1 to membrane interity and DNA damage etc, where similar graphics using a few compound concentrations are repeatedly presented. The naphthoquinones are said to be cytoprotective and are compared against idebenone, an orphan drug with unkown mechanism of action, suspected to bear antioxidant properties however, their clinical relevance being discussible. It is currently approved for the treatment of LHON.
The authors seem to acknowledge that naphthoquinones generate false results in most in vitro cell tests, therefore it is surprising for this reviewer to see this kind of investment in work related to this set of compounds. Both naphthoquinones and quinones are very well-known PAINs, i.e., pan-interfering assays. In addition, the comparison of naphothoquinones with ibedenone, that are quite clearly structurally unrelated especially when considering that there is no defined mode of action for idebenone that clearly explains its clinical efficacy, is not rational. Overall, to the best of this reviewer’s opinion, the study is devoid of meaningful content, has poor translation value, and there is heavy speculation in the results discussion.
Reviewer 3 Report
The authors present a compelling array of toxicity assays for their 11 experimental compounds. There are only two limitations to the study that should be noted by the authors. First, HepG2 cells are not as metabolically active as HepG2 C3A or even primary hepatocytes, so there is some limitation in the metabolic assay output. Second, the authors are targeting mitochondrial function and cytoprotection, yet not assessment of mitochondria is made by these assay. The addition of TMRE, mitoSOX, or even mitochondrial morphology MitoTracker or mito health markers such as pink, parkin, or mtDNA damage would enhance this analysis.
Two other general issues to consider. The physical growth characteristics of HepG2 cells make can have them grow into towers or “mini-organs” that create 3D structures that are harder to penetrate with dyes and drugs. Did the cells grow in this manner, and how would this more 3D like growth impact the penetrance of their drugs and their dye markers like the Multi-Tox. Finally, quinone structures can also act as endocrine disrupting agents, so future work may want to consider ER, AR, etc activity to minimize clinical side effects.
Minor issues:
Leigh syndrome listed twice in introduction